# COMPUTER-USE AGENTS AS JUDGES FOR AUTOMATIC GUI DESIGN

## ABSTRACT

Computer-Use Agents (CUA) are becoming increasingly capable of autonomously operating digital environments through Graphical User Interfaces (GUI). Yet, most GUI remain designed primarily for humans—prioritizing aesthetics and usability—forcing agents to adopt human-oriented behaviors that are unnecessary for efficient task execution. At the same time, rapid advances in coding-oriented language models (Coder) have transformed automatic GUI design. This raises a fundamental question: *Can CUA as judges to assist Coder for automatic GUI design*? To investigate, we introduce **AUI-Gym**, a benchmark for Automatic GUI development spanning 52 applications across diverse domains. Using language models, we synthesis 1560 tasks that simulate real-world scenarios. To ensure task reliability, we further develop a verifier that programmatically checks whether each task is executable within its environment. Building on this, we propose a **Coder–CUA in Collaboration** framework: the Coder acts as Designer, generating and revising websites, while the CUA serves as Judge, evaluating functionality and refining designs. Success is measured not by visual appearance, but by task solvability and CUA navigation success rate. To turn CUA feedback into usable guidance, we design an **CUA Dashboard** that compresses multi-step navigation histories into concise visual summaries, offering interpretable guidance for iterative redesign. By positioning agents as both designers and judges, our framework shifts interface design toward agent-native efficiency and reliability. Our work takes a step toward shifting agents from passive use toward active participation in digital environments.

## 1 INTRODUCTION

Recent advances in language agents have shown that **Computer-Use Agents** openai (2025); Anthropic (2025) can autonomously operate within GUIs—performing tasks such as online shopping by sequentially clicking through multiple buttons Zhou et al. (2024). However, today's environments remain fundamentally human-centric, optimized for aesthetics and usability through features like dynamic animations or colorful layouts. To adapt to these settings, researchers typically train CUA on large-scale human demonstration trajectories, click logs, or static screenshots Xu et al. (2024); Lin et al. (2024b); Seed (2025), effectively forcing agents to imitate human behavior. This approach binds automation to human-oriented design choices, where stylistic details crucial for humans are redundant for agents whose primary objective is efficient task completion. In parallel, coding-oriented language models—**Coders**—have already demonstrated strong capabilities, capable of generating functional HTML pages or even entire websites from a single instruction Si et al. (2024). Yet these outputs remain confined to human-facing loops: even when generated by agents, interfaces are still optimized for human use rather than agent-native interaction.

Both CUA and Coders thus exhibit remarkable potential for automation and design. This motivates a fundamental question: Can CUA assists Coders redesign UIs in an automatic manner—where environments are created for, and evaluated by, agents themselves, with CUA acting as judges? In this work, we reconceptualize the UI as a tunable environment. The core idea is to employ the *Coder as Designer*—responsible for initializing and revising UIs—while the *CUA acts as Judges*, navigating through tasks and collecting interaction trajectories as feedback.

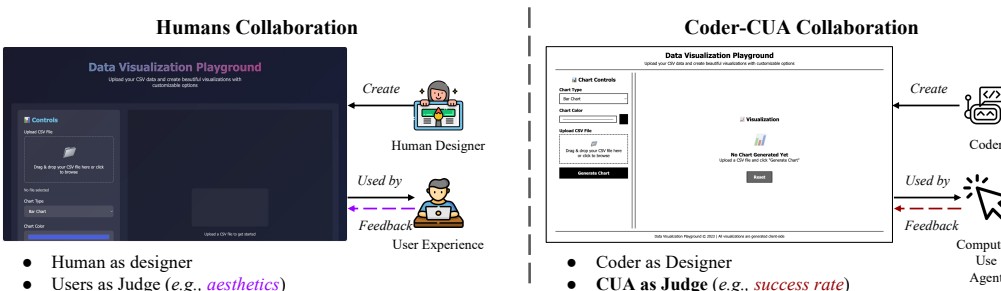

Figure 1: **Illustration of Humans Collaboration *vs.* our Coder-CUA Collaboration in term of UI designs. Left:** Most GUIs are designed by humans and optimized for user experience (*e.g.,* aesthetics), forcing trained agents to adapt to human-oriented behaviors. **Right:** Our Coder-CUA Collaboration framework leverages Coder as Designer and CUA as Judge together, enabling more reliable task execution and improved usability for agents.

As no existing testbed aligns with our goal, we introduce AUI-Gym to pioneer evaluation in this setting. AUI-Gym automatically develops websites across 52 applications spanning six domains (apps, landing pages, games, interactive demos, tools, and utilities). Unlike most coders that focus on single-page generation, AUI-Gym requires agents to produce fully automated, executable, application-level designs with an emphasis on functional completeness. Enabling sufficient, scalable, and human-free evaluation is non-trivial. however, is non-trivial. To simulate realistic usage scenarios, we prompt GPT-5 to propose 30 candidate tasks per application, yielding 1560 tasks in total. These tasks are then validated by humans. To ensure that each website can be reliably tested, GPT-5 also generates a customized rule-based functional checker for individual task, determining whether the task is feasible within the given interface. This infrastructure establishes a human-free, reliable foundation for subsequent CUA exploration and feedback-driven UI refinement.

To this end, we develop a **Coder–CUA collaboration** framework. The *Coder acts as Designer*, responsible for UI initialization and refinement, while the *CUA serves as Judge*, supplying feedback. The central challenge is how to transform raw CUA interactions into effective revision signals from an agent perspective. We address this through two complementary dimensions of feedback: **(a) CUA Navigation**, where the agent executes tasks through atomic actions such as clicks or typing and judges success or failure; and **(b) Task Solvability**, where unsolvable tasks are accumulated as functionality failures and returned to the Coder as precise indicators of missing features. CUA navigation produces long, multi-step trajectories interleaved with screenshots, making direct feedback difficult to interpret. To overcome this, we introduce the **CUA Dashboard**, which condenses each task, its outcome, actions, and intermediate states into a single $1920 \times 1080$ image. Rather than storing every screenshot, the dashboard highlights only key interactive regions, with region sizes adaptively scaled by the number of steps. This dynamic design reduces redundancy by average 76.2% while preserving essential cues, offering a clear step-by-step view of how the CUA perceives and acts on the interface. As a result, success and failure points become immediately visible, and the dashboard provides concise, interpretable feedback that the Coder to guide iterative UI redesign.

Our empirical results show that while state-of-the-art Coders can generate complete GUIs that appear suitable to humans, they still encounter notable limitations: **(i) Task solvability as a foundation.** Initial UIs often fail to capture many practical scenarios, resulting in low usability. However, by collecting failure cases, the Coder can readily boost performance through language-based functional summarization. **(ii) CUA navigation as a key bottleneck.** Even when UIs achieve high functional completeness, CUAs initially exhibit low success rates due to the complexity of multi-step navigation. Through our Coder–CUA collaboration, we substantially improve navigation success rates, particularly showing that CUA feedback-driven redesigns—such as *de-stylization*, increased contrast, and simplified layouts—significantly enhance CUA execution. Together, these findings highlight the promising potential of agents for automatic UI design and testing, improving both task success and robustness. To summarize, our contributions are threefold:

1. **AUI-Gym: a scalable testbed for automatic GUI development and testing**, covering 52 applications across six domains with 1560 GPT-5–proposed, human-validated tasks and per-task rule-based checkers. This enables human-free development of automatic UI creation and testing.

2. **Coder–CUA framework with CUA Dashboard.** The Coder initializes and refines UIs while the CUA judges via two signals: navigation outcomes and task solvability. A single-image $1K$

CUA Dashboard compresses task goal, actions, intermediate states, and outcome by highlighting key interactive regions with adaptive scaling, reducing visual tokens by 76.2% on average while preserving essential cues for redesign.

3. **Evaluation insights.** Task solvability is foundational yet readily improved via failure-driven functional summarization, whereas CUA navigation is the main bottleneck. Feedback-driven redesigns (*e.g.,* de-stylization, higher contrast, simplified layouts) substantially raise execution success and overall robustness.

## 2 RELATED WORKS

### 2.1 COMPUTER-USE AGENTS

Recent studies reveal the potential of LLMs beyond language modeling, with advancements in demonstrating their ability to autonomously complete complex tasks using tool integration Schick et al. (2023) like humans. This has prompted the development of GUI automation agents that learn to operate digital user interfaces by imitating human trajectories. This learning is primarily achieved in two ways: (i) by steering general multimodal foundation models Achiam et al. (2023) with in-context human trajectory examples, and the general models perceive the UI through intermediate representations like HTML, accessibility trees Drouin et al. (2024); Gao et al. (2023); Zheng et al. (2024), Optical Character Recognition Lu et al. (2024), or Set of Masks Yang et al. (2023). (ii) by pre-training specialized GUI foundation models through extensive supervised fine-tuning or reinforcement learning on large-scale vision-text UI data (*e.g.,* screenshots and instructions) Xu et al. (2024); Lin et al. (2024b); Gou et al. (2024); Lin et al. (2024a); Lu et al. (2025); Seed (2025). While foundational, these data-driven approaches suffered from heavy requirements for high-quality human trajectories to achieve agent performance improvements. Despite their methodological differences, these approaches share a common, agent-centric paradigm, focusing on improving the agent's capabilities to navigate a static and often complex environment. Notably, we investigate a complementary approach. Instead of adapting the agent, we explore how to dynamically tune the environment to enhance the performance of a frozen agent.

### 2.2 AUTOMATIC SOFTWARE DESIGNS

Besides CUAs, there have been extensive research on software automation, automatic interface design Lu et al. (2023); Kong et al. (2008) and generation Si et al. (2024); Beltramelli (2018); Laurençon et al. (2024). Programmatic and semantic UI components—such as accessibility layers, ARIA tags, and declarative interface frameworks (*e.g.,* React Native, Flutter)—illustrate how environments can be annotated or abstracted for automated processes. Similarly, benchmarks in automated software interaction, such as WebArena Zhou et al. (2024) and GAIA Mialon et al. (2023), assume agent operates within fixed, human-oriented systems for task automation. More recently, embodied AI environments (*e.g.,* ALFRED Shridhar et al. (2020), Habitat Puig et al. (2023), Mine-Dojo Fan et al. (2022)) show how environments can be crafted to accelerate agent training, though primarily in physical or simulated domains. These efforts highlight the growing recognition that environments themselves can be reimagined for machine interaction, yet a systematic framework for designing agent-centric digital environments in everyday computing remains absent.

## 3 AUI-GYM BENCHMARK

### 3.1 TASK DEFINITIONS

We develop AUI-Gym for automatic GUI development and testing. Given a language user query $\mathcal{Q}$ as input and several available agents (*e.g.,* Coder or CUA), the output is a complete website that serves as a *tunable* environment $\mathcal{E}$. We detail the input and output respectively below.

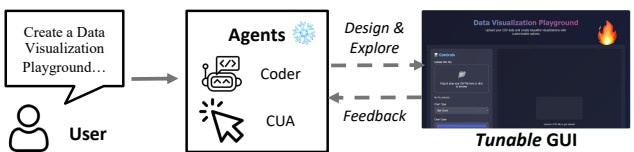

Figure 2: **AUI-Gym task definition.** A user issues a request (*e.g.,* "Create a Data Visualization Playground"), and agents (*e.g.,* Coder or CUA) interact with the GUI through design, exploration, and feedback. In this setup, the GUI serves as a *tunable* environment.

Figure 3: **AUI-Gym construction pipeline.** (i) An input query specifies the app requirements. (ii) GPT-5 proposes candidate tasks with explicit goals. (iii) Humans filter and refine tasks using domain-specific principles. (iv) A test-time Verifier reads the website HTML and generates task-specific, rule-based checkers to validate success on the to-be-tested website.

**Input Query** $\mathcal{Q}$**.** Since the outcome is a website, the user query $\mathcal{Q}$ should be both descriptive and concrete. To this end, we explicitly standardize queries into the structured format illustrated above. This supplements the query with a name, goal, functional features, and UI theme.

---

**Input formulation**

Create a single-page app in a single HTML file with the following requirements:
- Name: {Camping Gear Checklist}
- Goal: {Track gear for camping trips}.
- Features: {Checklist items, weight calculator, save lists.}
- Theme: {The UI should be outdoor-themed.}

---

**Output website** $\mathcal{E}$**.** The website is an application-level deliverable that must be fully functional, going beyond a static page to support navigation, transitions, button interactions, and completion of functional goals, with the objective of maximizing the agent's success rate. Constructing an effective evaluation framework in this setting is non-trivial and introduces several challenges. We next present our scalable, automatic solutions.

## 3.2 TASK CREATION

The full curation pipeline is illustrated in Fig.3. To construct the benchmark, we collect 52 task prompts from OpenAI's playground [1], covering multiple domains.

**Synthesize candidate tasks** $\mathcal{T}$**.** Applications are typically designed to support a variety of relevant tasks, and a key evaluation is whether they can smoothly handle such tasks. We leverage GPT-5 OpenAI (2025) to synthesize diverse user requirements: given an instruction $\mathcal{I}$, it generates a set of candidate tasks $\mathcal{T}$ that simulate practical usage. As illustrated in Fig. 3, for the application '*Micro Habit Tracker*', an example task is: "*Create a habit named 'Meditate 5 min,' then view today's column and the habit chart.*" These tasks serve as fine-grained probes that capture the potential demands of the environment $\mathcal{E}$.

**Manual quality control.** As the tasks are automatically generated by GPT-5, human oversight is required to ensure their quality. Different applications demand different characteristics: for example, tasks for game UIs should emphasize interactivity and control, while tasks for utility tools should capture information accessibility and workflow patterns. To this end, humans define domain-specific principles and filter out low-quality tasks (*e.g.,* trivial clicks) or nonsensical ones (*e.g.,* beyond the application scope), ambiguous query (cross-application), ensuring that the proposed tasks remain concrete, meaningful and aligned with each domain's design philosophy.

**Data Statistics.** Based on the above strategy, we obtain 30 tasks for each application. The benchmark spans 52 web applications across six domains, yielding a total of 1,560 tasks and enabling comprehensive evaluation across diverse applications. As illustrated in Table 7, the domains include: (i) App, general-purpose applications; (ii) Landing, commercial and promotional interfaces; (iii) Game, puzzle and arcade-style challenges; (iv) Interactive, dynamic user engagement with real-time feedback; (v) Tool, specialized utilities; and (vi) Utility, everyday organizational support. This diverse coverage captures distinct GUI challenges—ensuring robust evaluation across varied interaction paradigms and functional complexities.

---

[1]https://github.com/openai/gpt-5-coding-examples

Table 1: **Examples of App domains in AUI-Gym.** For each domain, we show a website created by GPT-5, paired with 30 tasks (blue) simulating real-world usage. Each task is further linked to a rule-based verifier (green). See full distribution and examples in Tab.7.

| Domain | #Apps | Percentage | Example Instruction | GUI created by GPT-5 |
|---|---|---|---|---|
| App | 11 | 21% | Create a single-page app in a single HTML file with the following requirements:
- Name: Healthy Meal Tracker
- Goal: Log meals and nutrition info.
- Features: Ingredient list, calories per meal, daily summary.
- The UI should be clean with food icons.
**Task: Add five meals for today's date (any names/ingredients) so today's meal count reaches at least 5.**
**Rule: `#dailyMealCount >= 5`** |  |

## 3.3 Evaluation with Verifiers

Even with the proposed tasks, it remains challenging to determine whether a given GUI can truly satisfy them, as websites are interactive and highly diverse environments. More importantly, since the *GUIs are generated at test time*, it is difficult to design fixed standards that generalize across all cases, given the variety of possible implementation approaches. A naive solution is to adopt a VLM-as-Judge approach, but this inevitably introduces bias and uncertainty. Ideally, the most reliable solution would be concrete functional checks with manual validation, yet this approach is prohibitively expensive and labor-intensive.

To address this, we define a **Verifier** $\mathcal{V}(\cdot)$ powered by GPT-5 at test time, which takes as input a candidate GUI together with a specific task. It analyzes the available elements and states, reasoning over the presence of required UI components,

```
Verifier(input = GUI_HTML, task):
    analyze elements and states
    if task solvable:
        return (Yes, function_checker)
    else:
        return (No, None)
```

their properties, and potential interaction paths. If the task is deemed solvable, the Verifier produces a task-specific verification **function checker** $\widetilde{\mathcal{V}}(\cdot)$ (by `JavaScript`) that encodes the success condition by element status; otherwise, the task is discarded as invalid, preventing noisy or unachievable goals from disrupting evaluation. Such as in Fig.3, for task "*Create a habit named 'Meditate 5 min,' then view today's column and the habit chart.*", based on the candidate website (right), the verifier generate the rule `gridContainer contains 'Meditate 5min'` In this way, the Verifier is customized for each website and each task at test time, ensuring reliable validation.

**Metrics.** With the support of function checkers as reliable verification, we can ensure that a website is both actionable and workable for the CUA. This further allows us to evaluate whether tasks are completed after CUA navigation, thereby measuring task success rate within the UI environment. In this way, we divise the following measure:

**(i) CUA Success Rate (SR).** This measures the average success rate over all tasks executed by CUA. If CUA successfully completes a task, it is counted as a success; otherwise, it is counted as a failure. Notably, if the Coder fails to yield a functional checker, the task is counted as a failure.

$$\text{SR} = \frac{1}{|\mathcal{T}|} \sum_{t \in \mathcal{T}} \mathbf{1} \left( \text{task } t \text{ is successfully completed} \right), \tag{1}$$

where $\mathcal{T}$ denotes the set of all tasks and $\mathbf{1}\{\cdot\}$ is the indicator function.

**(ii) Function Completeness (FC).** While CUA performance reflects the ultimate goal, it may be sparse if most CUAs fail to complete tasks. Therefore, we devise a second metric to evaluate only whether the Coder-created website functionally supports the task (valid), independent of CUA navigation. This metric reflects task validity and serves as a more basic measure.

$$\text{FC} = \frac{1}{|\mathcal{T}|} \sum_{t \in \mathcal{T}} \mathbf{1}\{\text{a functional checker exists for task } t\}. \tag{2}$$

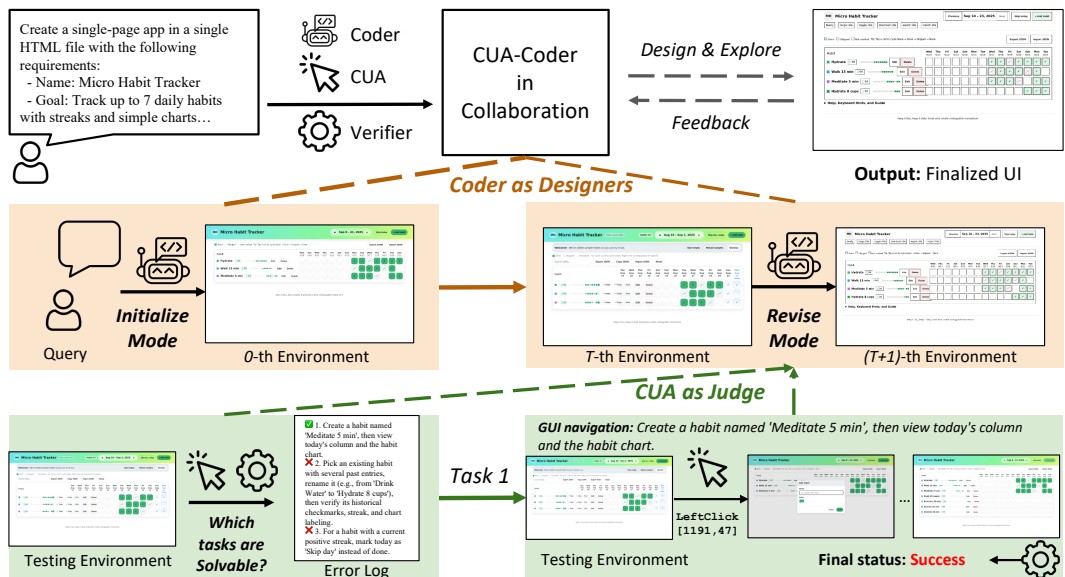

Figure 4: **Overview of the Agent-centric User Interface (AUI) framework**. The process begins with the Coder as Designer, which initializes and iteratively revises the UI based on queries and feedback. In parallel, the CUA as Judge executes task-driven navigation within the testing environment, generating trajectories and error logs to evaluate task solvability. A verifier ensures functional correctness, while feedback from CUA navigation informs subsequent UI revisions. This collaboration yields a finalized agent-centric UI optimized for both functionality and execution success.

## 4    CUA–CODER IN COLLABORATION

**Overview.** We present our framework for enabling collaboration between the CUA and the Coder, consisting of two main components: the Coder as Designer while the CUA as Judge. Given a user instruction $\mathcal{Q}$, AUI generates an initial UI environment $\mathcal{E}_0$, which is iteratively revised through interaction and feedback. The framework involves two central roles: a Coder policy $\pi_{\text{Coder}}$ that proposes and revises UI designs, and a CUA policy $\pi_{\text{CUA}}$ that explores the UI and evaluates its functionality. We formalize this process as a Markov Design Process. The state is the current UI $\mathcal{E}_t$, the action is a design update proposed by $\pi_{\text{Coder}}$, and the transition deterministically $\mathcal{E}_{t+1} \leftarrow \pi_{\text{Coder}}(\mathcal{E}_t, \mathcal{R}_t)$. The feedback $\mathcal{R}_t$ is relate to the metrics (*i.e.,* Eq.1 and Eq.2) results achieved by the CUA when interacting with $\mathcal{E}_t$, *i.e.,* $\mathcal{R}_t \leftarrow S(\mathcal{E}_t, \pi_{\text{CUA}})$. The Coder is optimized to maximize the total reward $\mathbb{E}\left[\sum_t \gamma^t \mathcal{R}_t\right]$. In this formulation, the CUA acts as a user that provides actionable feedback by testing the environment, while the Coder serves as a designer who integrates this feedback into code revisions to iteratively improve the UI. Unlike conventional CUA setups, where the agent adapts to a fixed environment $\pi_{\text{CUA}} \leftarrow \mathcal{E}$, our framework adapts the environment itself based on CUA feedback $\mathcal{E} \leftarrow \pi_{\text{CUA}}$, thereby optimizing UIs for agent-native success. We illustrate the full workflow in Fig. 4 and detail each role in the following subsections.

### 4.1    CODER AS DESIGNERS

Recent advances in Coder OpenAI (2025); Qwen (2025); anthropic (2025) demonstrate strong capabilities in generating UI applications. In our framework, we position Coders as *designers*, responsible not only for creating new environments but also for refining them based on feedback from CUAs. Accordingly, Coders operate in two complementary modes: one dedicated to the initial creation of UIs, and the other focused on their iterative improvement through CUA-guided feedback.

**i. Initialization.** Given a user query defined in formulation 3.1 and enriched with multiple details, the Coder progressively generates long-context code to construct a complete HTML-rendered UI $\mathcal{E}_0$ from scratch, which serves as the base environment for subsequent interactions.

| Task | Load the app for the first time and wait for the curtain reveal to complete. |
|---|---|
| Website | 
$1280 \times 720$ |
| Dashboard
(an image) | 
Before: $6 \times 1280 \times 720 \rightarrow$ After: $1 \times 1950 \times 975$, 76.2% tokens reduction |
| Result | Failure |
| Comments | The weather-theatre app requires button clicks to trigger curtain reveal, but the task expects automatic curtain opening on first load without user interaction, creating a fundamental mismatch between expected auto-start behavior and actual manual activation requirement. |

Table 2: **Illustration of CUA Dashboard.** The dashboard generates one informative image that clearly demonstrates how the CUA performs each step along with the corresponding observations, while reducing visual tokens by cropping to the key interactive regions.

**ii. Revision from Feedback.** After constructing the initial environment $\mathcal{E}_0$, the Coder enters an iterative refinement loop to update the UI: $\mathcal{E}_{t+1} \leftarrow (\mathcal{E}_t, \mathcal{R}_t)$, where $\mathcal{R}_t$ denotes the feedback signal expressed as a language caption, described in the next section.

## 4.2 CUA AS JUDGES

We employ Computer-Use Agents (CUAs) as *Judges* to trial and diagnose the UIs $\mathcal{E}_t$ generated by the Coder, providing actionable feedback for iterative redesign. Specifically, we define two complementary forms of reward signals:

**(i) Task Solvability Feedback $\mathcal{R}_{\text{task}}$.** Before navigation begins, we verify whether a task $\tau$ is implementable on the current UI. Let $\mathcal{V}$ denote the verifier in Sec. 3.3. A task is deemed solvable if and only if $\mathcal{V}(\mathcal{E}_t, \tau) = 1$; otherwise it is labeled a *functionality failure*. This gate prevents wasted rollouts on impossible tasks and sharpens the feedback signal. We collect all failed tasks into $\mathcal{T}_{\text{fail}} = \{\tau : \mathcal{V}(\mathcal{E}_t, \tau) = 0\}$ and return them to the Coder as precise indicators of missing features. The Coder then aggregates and summarizes these failures into a language feedback signal $\mathcal{R}_{\text{task}}$.

**(ii) CUA Navigation Feedback $\mathcal{R}_{\text{nav}}$.** For solvable tasks $\mathcal{T}_{\text{succ}} = \{\tau : \mathcal{V}(\mathcal{E}_t, \tau) = 1\}$, evaluation proceeds as a UI navigation problem. At step $k$, the CUA receives an observation $o_k$ (a screenshot of the current state), emits an action $a_k \in \{\text{CLICK}, \text{TYPE}, \text{SCROLL}, \ldots\}$ with an optional reasoning trace, and the environment transitions to the next state, yielding $o_{k+1}$. The trajectory

Table 3: **Main results on AUI-Gym per Coder.** Top: Function Completeness Rate (%). Bottom: CUA Success Rate (%).

| Coder | Feedback Type | landing (%) | game (%) | app (%) | utility (%) | interactive (%) | tool (%) | overall (%) |
|---|---|---|---|---|---|---|---|---|
| | | | | *Function Completeness* | | | | |
| GPT-5 | Baseline | 53.0 | 77.8 | 70.6 | 63.3 | 73.0 | 70.0 | 67.9 |
| | + Task Solvability | 19.7 | 100.0 | 69.4 | 65.6 | 55.6 | 56.2 | 60.5 |
| | + CUA Navigation | 53.3 | 87.8 | 74.2 | 70.0 | 70.4 | 69.5 | 70.8 |
| | + Integrated | 75.3 | 92.2 | 85.2 | 73.3 | 82.6 | 76.7 | **81.5** |
| Qwen3-Coder-30B | Baseline | 16.3 | 50.4 | 41.2 | 43.9 | 52.2 | 54.8 | 42.1 |
| | + Task Solvability | 55.0 | 79.6 | 58.5 | 67.8 | 56.3 | 74.3 | **64.3** |
| | + CUA Navigation | 23.3 | 50.4 | 38.8 | 49.4 | 39.3 | 55.2 | 41.3 |
| | + Integrated | 47.7 | 72.2 | 59.7 | 56.7 | 57.0 | 69.5 | 60.1 |
| GPT-4o | Baseline | 9.7 | 55.2 | 36.1 | 38.9 | 44.8 | 37.6 | 36.3 |
| | + Task Solvability | 23.7 | 55.9 | 52.1 | 55.0 | 58.9 | 65.2 | **50.6** |
| | + CUA Navigation | 8.3 | 55.2 | 28.2 | 34.4 | 26.3 | 35.7 | 30.4 |
| | + Integrated | 16.3 | 68.5 | 36.4 | 51.7 | 51.1 | 41.4 | 43.1 |
| | | | | *CUA Success Rate* | | | | |
| GPT-5 | Baseline | 34.7 | 24.8 | 27.3 | 14.4 | 18.1 | 21.9 | 24.5 |
| | + Task Solvability | 16.3 | 39.3 | 26.7 | 16.1 | 20.7 | 11.9 | 22.6 |
| | + CUA Navigation | 17.7 | 43.3 | 30.0 | 21.1 | 21.1 | 17.6 | 25.7 |
| | + Integrated | 40.7 | 27.4 | 31.5 | 22.2 | 14.1 | 12.9 | **26.0** |
| Qwen3-Coder-30B | Baseline | 5.3 | 9.3 | 9.1 | 11.7 | 7.0 | 1.4 | 7.3 |
| | + Task Solvability | 14.7 | 42.2 | 19.1 | 14.4 | 11.1 | 4.3 | 18.3 |
| | + CUA Navigation | 6.7 | 20.7 | 9.1 | 11.1 | 12.2 | 11.4 | 11.7 |
| | + Integrated | 23.7 | 30.7 | 22.4 | 7.8 | 9.3 | 13.8 | **19.0** |
| GPT-4o | Baseline | 4.7 | 12.6 | 12.4 | 6.7 | 9.3 | 5.7 | 8.8 |
| | + Task Solvability | 8.7 | 18.5 | 19.1 | 5.6 | 8.5 | 22.9 | 14.1 |
| | + CUA Navigation | 5.7 | 31.5 | 10.0 | 8.3 | 10.4 | 6.7 | 12.3 |
| | + Integrated | 10.3 | 27.4 | 13.9 | 13.3 | 15.2 | 16.7 | **16.1** |

$\mathcal{H} = (o_0, a_0, \ldots, o_K)$ terminates when either (a) the function checker signals success $\widetilde{\mathcal{V}}(\mathcal{E}_t, \tau) = 1$, or (b) a step limit is reached, which we record as a failure. We log full trajectories—observations, actions, and intermediate rationales—and use them to construct targeted feedback for UI refinement.

**CUA Dashboard for Compact Feedback.** Raw trajectories $\mathcal{H}$ are long and interleaved, making them ill-suited for direct ingestion by the Coder. We therefore distill each rollout into an CUA Dashboard (Fig. 2): a single, fixed-resolution ($1920 \times 1080$) canvas that compresses key evidence from the trial. Rather than storing full frames, we crop and tile only *interactive regions* touched by the CUA, allocating dynamic region sizes based on step order to preserve temporal structure. This yields a substantial reduction in redundancy (*e.g.,* a 76.2% drop in visual content) while retaining the cues needed to localize failure modes (missed affordances, hidden state, ambiguous labels) and success paths at a glance. The dashboard provides a step-by-step visual trace aligned with actions, making error locations immediately visible. Finally, we convert the dashboard into a concise language summary $\mathcal{R}_{nav}$ by passing it to the Coder and as the feedback used in the revision rule.

## 5 EXPERIMENTS

### 5.1 SETTINGS

**Baselines.** For the Coder, we select representative models including GPT-5 OpenAI (2025), GPT-4o Hurst et al. (2024), and the open-source Qwen3-Coder Qwen (2025). For the CUA, we adopt UI-TARS-1.5-7B Seed (2025) and Operator openai (2025). UI-TARS-1.5-7B is a lightweight yet strong performer among open models with high efficiency, and Operator is among the state-of-the-art close-source API-based CUAs. The experiment results highlight that the performance gain brought by our proposed method is universal for both lightweight and powerful CUAs.

### 5.2 MAIN RESULTS

Table 3 reports results across six domains for three coders. Several key findings emerge: **(i) Function Completeness.** Revision based on task solvability feedback leads to substantial gains, consistently boosting the overall functionality completeness for all coders. After applying integrated

revision for GPT-5, the function completeness is increased to 81.5% from 67.9%, reaching the highest. Notably, the landing, game and app domains have dramatic improvements, with the maximum improvements of 38.7%. Interestingly, Revision based on task solvability feedback or CUA navigation feedback alone does not guarantee function completeness improvents, but the integrated revision combining these two components bring stable improvents on all domains for all coders, highlighting the strength of our design. Moreover, fixing unresolved functionalities alone also benefits CUA task solving, yielding a 4.8% average improvement on CUA evaluation, highlighting the mutual reinforcement between task solvability and CUA navigation.

**(ii) CUA performance.** Open-source CUAs initially perform poorly, with an average overall CUA success rate of only 13.5%. However, our framework can consistently improve the CUA success rate, with an average 6.8% improvements. Interestingly, our framework bring large improvements to weak coders such as Qwen3-Coder-30B and GPT-4o, with a maximum overall improvement of 11.7%, showcasing that our framework can greatly empower weak models. Overall, these results demonstrate the effectiveness of our framework: task solvability feedback guides to robust UI design, while leveraging CUA navigation feedback optimizes interfaces toward agent-centric success.

## 6 KEY ABLATIONS

**Effects by different CUAs choices.** In Fig. 5, we compare UI-TARS and Operator as CUA policies within the integrated revision loop. We evaluate with two coders—GPT-5 (closed-source, stronger) and Qwen3-Coder-30B (open-source, weaker)—to cover both capability and licensing spectra. Both CUA policies yield comparable gains in functional completeness, with UI-TARS slightly outperforming on Qwen3-Coder-30B. Although the task-solvability signal is identical across CUAs, UI-TARS tends to fail more tasks, thereby surfacing richer failure cases and driving greater function-oriented revisions. For CUA success rate (SR), Operator delivers larger gains with the stronger coder (GPT-5), while improvements are similar across CUAs for the weaker coder. This suggests Operator's navigation strengths are best realized on more complex UIs, whereas weaker coders often produce simpler interfaces. Overall, lightweight open-source CUAs like UI-TARS are an efficient and effective choice for harvesting navigation feedback in practice.

See the Appendix and for more ablations (Sec. A) and quantification examples (Sec. E).

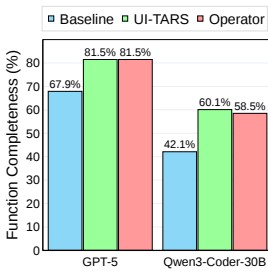
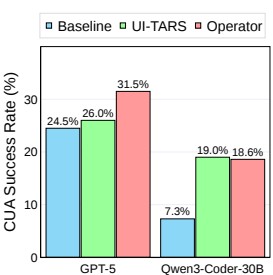

| (a) Function completeness. | (b) CUA success rate. |

Figure 5: **Effect by different Coders**, including open-source UI-TARS and SoTA Operator.

## 7 CONCLUSION

We introduced AUI-Gym, a new benchmark for automatic GUI development (52 applications; 1,560 tasks with programmatic checkers), and a Coder–CUA collaboration framework that recasts UI design as an agent-native loop, with the Coder as Designer and the CUA as Judge. Central to this loop is the CUA Dashboard, which compresses long navigation trajectories into compact, interpretable summaries that convert raw interactions into actionable revision signals. Empirically, task solvability is foundational—readily improved by failure-driven functional summarization—whereas CUA navigation remains the primary bottleneck; feedback-driven redesigns (*e.g.,* de-stylization, higher contrast, simplified layouts) consistently raise execution success and robustness, highlighting the value of designing *for* agents rather than merely adapting human-centric interfaces.

ETHICS STATEMENT

This work proposes a UI generation benchmark AUI-Gym, a multi-agent framework AUI and does not involve sensitive or private information. Human annotation was conducted with informed consent and fair compensation. We see minimal risk of harm; potential misuse (e.g., generating misleading visualizations) is noted, and we release our benchmark strictly for research purposes.

REPRODUCIBILITY STATEMENT

We provide details of dataset construction, evaluation protocols, and model settings in the main text and appendix. All data used are publicly available, and our benchmark, code, and evaluation scripts will be released upon publication to facilitate replication of our results.

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

## A  EXPERIMENT RESULTS

Below, we show the experiment results additional to the results showcased in the main paper.

Table 4: **Main results per model (Operator as CUA)**: CUA Success Rate (Function Completeness Rate).

| Model | Version | landing | game | app | utility | interactive | tool | overall |
|---|---|---|---|---|---|---|---|---|
| GPT-5 | Baseline | 34.7% (53.0%) | 24.8% (77.8%) | 27.3% (70.6%) | 14.4% (63.3%) | 18.1% (73.0%) | 21.9% (70.0%) | 24.5% (67.9%) |
| | + Integrated | 41.3% (75.3%) | 42.6% (92.2%) | 38.8% (85.2%) | 27.8% (73.3%) | 10.7% (82.6%) | 21.4% (76.7%) | **31.5% (81.5%)** |
| Qwen3-Coder-30B | Baseline | 5.3% (16.3%) | 9.3% (50.4%) | 9.1% (41.2%) | 11.7% (43.9%) | 7.0% (52.2%) | 1.4% (54.8%) | 7.3% (42.1%) |
| | + Integrated | 10.0% (47.0%) | 27.0% (68.9%) | 19.1% (60.3%) | 20.6% (55.6%) | 13.7% (57.4%) | 23.8% (62.9%) | **18.6% (58.5%)** |
| GPT-4o | Baseline | 4.7% (9.7%) | 12.6% (55.2%) | 12.4% (36.1%) | 6.7% (38.9%) | 9.3% (44.8%) | 5.7% (37.6%) | 8.8% (36.3%) |
| | + Integrated | 15.7% (19.0%) | 35.9% (59.3%) | 14.5% (44.5%) | 15.0% (47.8%) | 5.9% (50.7%) | 13.8% (46.2%) | **16.9% (43.8%)** |

As shown in the Table 4, when using operator as CUA policy for integrated revision, consistent improvements for both function completeness and CUA success rate can be observed. Moreover, compared to the CUA success rate showcased in Table 3, it can be observed that Operator has higher CUA success rate than UI-TARS in hard domains such as game and app that requires responsive and complex intractions, showcasing its strong navigation capability.

Table 5: **Commenter Ablations per model**: CUA Success Rate (Function Completeness Rate).

| Model | Variant | landing | game | app | utility | interactive | tool | overall |
|---|---|---|---|---|---|---|---|---|
| GPT-5 | Text-only | 24.0% (50.7%) | 31.2% (87.8%) | 21.2% (69.4%) | 16.1% (55.6%) | 8.9% (59.3%) | 6.2% (43.3%) | 18.7% (62.1%) |
| | Screenshot-only | 17.3% (30.3%) | 16.7% (65.6%) | 12.4% (42.7%) | 15.6% (38.3%) | 5.2% (27.8%) | 9.5% (46.7%) | 12.8% (41.7%) |
| | Dashboard | 17.7% (53.3%) | 43.3% (87.8%) | 30.0% (74.2%) | 21.1% (70.0%) | 21.1% (70.4%) | 17.6% (69.5%) | **25.7% (70.8%)** |
| Qwen3-Coder-30B | Text-only | 8.0% (18.3%) | 20.7% (61.9%) | 7.3% (42.4%) | 8.3% (54.4%) | 10.7% (48.9%) | 16.2% (57.1%) | **11.7% (45.6%)** |
| | Screenshot-only | 9.3% (20.7%) | 11.9% (63.7%) | 5.2% (34.5%) | 10.6% (40.6%) | 7.4% (55.9%) | 5.2% (37.6%) | 8.1% (41.7%) |
| | Dashboard | 6.7% (23.3%) | 20.7% (50.4%) | 9.1% (38.8%) | 11.1% (49.4%) | 12.2% (39.3%) | 11.4% (55.2%) | **11.7% (41.3%)** |
| GPT-4o | Text-only | 7.7% (13.0%) | 14.8% (57.0%) | 12.7% (34.8%) | 2.8% (37.8%) | 15.9% (39.3%) | 7.6% (29.0%) | 10.8% (34.8%) |
| | Screenshot-only | 4.7% (10.3%) | 15.6% (43.7%) | 10.6% (31.2%) | 6.1% (45.6%) | 5.6% (34.8%) | 7.1% (37.6%) | 8.5% (32.5%) |
| | Dashboard | 5.7% (8.3%) | 31.5% (55.2%) | 10.0% (28.2%) | 8.3% (34.4%) | 10.4% (26.3%) | 6.7% (35.7%) | **12.3% (30.4%)** |

Table 5 demonstrates the results when using different types of commenters for revision based on CUA navigation feedback. From the results, it can be inferred that dashboard is capable to provide comprehensive visual and textual cues derived from the CUA policy trajectories, but requiring the commenter to have strong visual perception.

Table 6 demonstrates that why VLM evaluation on CUA task trajectory is unreliable. It can be observed that the the compared to rule-based oracle evaluation, the VLM evalation tends to judge the CUA policy outcome as failure, thus having very low balanced accuracy, recall and F1. Moreover, the low Cohen's $\kappa$ indicates very weak agreement of VLM evaluation compared to rule-based oracle evaluation. This indicates that VLM evaluation on the final screenshot only is unreliable, and may requires more screenshots along the CUA policy task trajectory for more reliable evaluation, leading to much higher computational cost.

Table 6: **VLM evaluation on final screenshot only is unreliable.** Given the final screenshot of CUA trajectory and the expected outcome, the accuracy of VLM evaluation is only slightly above the naive all-fail baseline; Balanced Accuracy is near 0.55; Recall/F1 and Cohen's $\kappa$ are low.

| Metric | Overall | GPT-5 | Qwen2.5-VL-72B | GPT-4o |
|---|---|---|---|---|
| Naive all-fail baseline accuracy | 0.720 | 0.720 | 0.720 | 0.720 |
| Accuracy vs. oracle | 0.735 | 0.736 | 0.738 | 0.732 |
| Balanced accuracy | 0.556 | 0.549 | 0.568 | 0.552 |
| Precision (Pass) | 0.616 | 0.660 | 0.612 | 0.589 |
| Recall (Pass) | 0.147 | 0.121 | 0.178 | 0.142 |
| F1 (Pass) | 0.237 | 0.205 | 0.276 | 0.229 |
| Cohen's $\kappa$ | 0.145 | 0.128 | 0.175 | 0.132 |

## B  AGENT'S PROMPTS

**Task Proposer Prompt.** Below, we show the prompt used by the Task Proposer.

---

**Task Proposer Prompt**

Propose a comprehensive set of 30 diverse, realistic user tasks for the following {tag_type} application:

Application: {app_title}
Description: {app_description}

Each task should be:
- Clear and specific in its description
- Represent realistic user scenarios
- Cover different complexity levels and use cases
- Grounded in an observable outcome: The task's completion must be marked by a clear and unambiguous change in the application's state or interface. The expected outcome description must precisely define this terminal state.
- Avoid single element grounding (focus on complete workflows)
- Test the application's core functionality effectively

{tag_specific_content}

Tag Philosophy Template:

game:
Focus on GAME-SPECIFIC user tasks:
1. Playing complete game rounds or levels
2. Achieving high scores and personal bests
3. Completing specific game objectives or challenges
4. Using game controls and input methods
5. Navigating game menus and settings
6. Restarting games and trying different strategies
7. Progressing through difficulty levels

Additional task requirements:
- Focus on actual gameplay actions and goals
- Include winning and losing scenarios
- Cover different skill levels and strategies
- Test game restart and replay functionality
- Emphasize user enjoyment and engagement

tool:
Focus on TOOL-SPECIFIC user tasks:
1. Creating or generating content using the tool
2. Inputting data in various formats and types (typed/pasted text or on-page controls)

---

3. Transforming and processing information
4. Previewing results in-page (no file uploads/downloads)
5. Using tool-specific features and options
6. Working with both simple and complex inputs
7. Completing end-to-end workflows within the page

Additional task requirements:
- Focus on practical use cases and workflows
- Include both basic and advanced tool usage
- Cover different input types and scenarios without external files
- Verify visible in-page outputs or status changes in the DOM
- Emphasize real-world problem solving

utility:
Focus on UTILITY-SPECIFIC user tasks:
1. Setting up and configuring the utility for personal use
2. Adding, organizing, and managing data or items
3. Tracking progress and monitoring status over time
4. Using timers, reminders, and scheduling features
5. Customizing settings and preferences
6. Completing daily or routine activities
7. Accessing and updating information quickly

Additional task requirements:
- Focus on everyday productivity scenarios
- Include setup and personalization tasks
- Cover routine and habitual usage patterns
- Test organization and tracking features
- Emphasize practical daily life applications

interactive:
Focus on INTERACTIVE-SPECIFIC user tasks:
1. Exploring and experimenting with interactive elements
2. Creating and manipulating visual or audio content
3. Adjusting parameters and settings in real-time
4. Playing with creative tools and features
5. Experiencing immersive visual or audio effects
6. Using touch, click, and gesture interactions
7. Customizing appearance and behavior

Additional task requirements:
- Focus on creative and exploratory activities
- Include experimentation and play scenarios
- Cover different interaction methods
- Test customization and personalization
- Emphasize sensory and aesthetic experiences

landing:
Focus on LANDING-SPECIFIC user tasks:
1. Browsing and exploring page content and sections
2. Reading and understanding key information
3. Clicking on call-to-action buttons and links
4. Navigating through different page sections
5. Finding contact information and ways to engage
6. Viewing team, product, or service details
7. Accessing additional resources and links

Additional task requirements:
- Focus on visitor browsing and exploration
- Include information-seeking behaviors
- Cover engagement and conversion actions
- Test navigation and content discovery
- Emphasize typical visitor journey scenarios

app (default/other):
Focus on APP-SPECIFIC user tasks:
1. Creating, editing, and managing content or data
2. Using multiple features in combination
3. Setting up and personalizing the application
4. Completing complex multi-step workflows
5. Organizing and categorizing information
6. Accessing and updating saved information

Additional task requirements:
- Focus on practical in-app usage
- Include multi-feature workflows and combinations
- Cover content creation and management
- Test personalization and customization
- Verify completion via visible state changes in the DOM (no external integrations)

Task Categorization Framework:
Each task must be categorized into one of the following three archetypes, which provides a structured approach to evaluating different facets of the application's functionality:
- "core_function": Tests a single, primary feature in isolation.
- "user_workflow": Tests a sequence of features that represent a complete user goal.
- "edge_case": Tests non-standard inputs, boundary conditions, or less common interaction patterns.

Please respond in JSON format:

```
{
  "app_name": "<app_name>",
  "tags": ["<tag1>", "..."],
  "tasks": [
    {
      "id": 1,
      "description": "Clear, specific task description",
      "category": "core_function|user_workflow|edge_case",
      "expected_outcome": "What should happen when task completes"
    }
  ]
}
```

**Coder Prompt.** Below, we show the prompt used by the Coder.

### Coder Prompt

[Initial Website Generation]

Create a single-page web application based on the following specification:

{instruction}

Requirements:
1. Create a complete HTML file with embedded CSS and JavaScript
2. The app should be fully functional and interactive
3. Use modern HTML5, CSS3, and vanilla JavaScript (no external libraries)
4. Include proper semantic HTML structure
5. Make the UI clean, responsive, and user-friendly
6. Add unique IDs to interactive elements for easier automation testing
7. Ensure the app works in a 1280x720 viewport

Please generate the complete HTML file:

[Revision from CUA Failures — Core Prompt]

You are tasked with improving a web application based on detailed failure analysis from automated testing.

## CONTEXT
Application: {app_name}
Model: {model_name}
Total Failed Tasks: {len(failed_tasks)}
Failure Categories: {list(failure_categories.keys())}
Original HTML Length: {len(initial_html.strip())}

## OUTPUT FORMAT
Generate a single, complete, and self-contained HTML file. The file must be fully functional, including all necessary CSS and JavaScript, from '!DOCTYPE html' to '/html'. Do not use placeholders or truncate the code.

## ORIGINAL INITIAL WEBSITE (FULL)

{initial_html}
 ## COMMENTER UI ANALYSIS
{(failure_analysis or "No visual UI analysis available").strip()}

{(non_regression_contract_prompt or ").strip()}

## IMPROVEMENT REQUIREMENTS

### 1. Core Issues to Address
Based on the failure analysis, you must:
- Identify missing DOM elements that tasks expect to exist
- Add missing JavaScript functionality for user interactions
- Fix timing issues that prevent task completion
- Ensure proper event handling and state management
- Add missing visual feedback and UI updates

### 2. Specific Fixes Needed
For each failed task category:
- **basic_usage**: Ensure fundamental interactions work (clicking, displaying, updating)
- **workflow**: Support complete user workflows and multi-step processes
- **advanced_feature**: Implement sophisticated UI behaviors and animations
- **edge_case**: Handle unusual inputs and boundary conditions properly

### 3. Technical Implementation Guidelines
- Preserve ALL existing working functionality from initial version
- Add missing HTML elements with unique IDs for automation
- Implement complete JavaScript event handlers and state updates
- Ensure synchronous UI updates for immediate feedback
- Do NOT introduce new input constraints that would block task inputs implied by the tasks (e.g., accept plain text or non-HTTP payloads if tasks need them). Validation must be permissive and never reduce what the initial version allowed.
- Do NOT auto-trigger flows on page load that would change initial states relied upon by tasks (e.g., auto-generation, auto-download, auto-navigation). Initial state should be neutral and idle.
- Keep critical controls visible within a 1280x720 viewport without scrolling. Avoid multi-panel "hub" layouts; prefer single-view, compact layouts that fit important controls on screen.
- Avoid adding non-essential animations/transitions; prioritize high visibility and clarity over decoration.
- Make sure timers, counters, and dynamic content work correctly

### 4. DOM Structure Requirements
- Every interactive element MUST have a unique ID
- Form controls must have proper event listeners
- Dynamic content areas must update immediately on state changes
- Visual feedback must be implemented for all user actions

### 5. JavaScript Functionality Requirements

- All user interactions mentioned in failed tasks must be fully implemented
- State changes must be reflected in the DOM immediately
- Event handlers must properly update all related UI elements
- Any game logic, scoring, timing must be complete and functional

Surgical Revision Policy
- Preserve existing IDs; do not rename or remove working elements from initial version.
- Avoid large rewrites. Patch only the functions, event handlers, and minimal markup necessary to satisfy the failed/unsupported tasks.
- Preserve working logic from initial version; do not regress features that already work.
- Reuse existing elements/IDs for state wherever possible; only add new IDs if strictly necessary to expose the state of new logic.
- Preserve initial version immediacy semantics. Do NOT introduce extra confirmation steps as prerequisites where initial version achieved completion via immediate interactions. Implement functional logic first, then expose proxies from the same code path; never update proxies without the underlying state change.

Commenter JSON (if provided)
- If the COMMENTER UI ANALYSIS is a JSON object, prioritize applying entries in 'actionable_changes' precisely.
- Keep changes surgical and bounded by those actionable suggestions; do not broaden scope beyond them.

## OUTPUT REQUIREMENTS
Generate a COMPLETE, FULLY FUNCTIONAL HTML file that:
1. Addresses ALL failure points identified in the analysis
2. Maintains existing successful functionality from initial version
3. Implements missing features causing task failures
4. Provides proper DOM elements for automation testing
5. Ensures immediate UI feedback for all user actions

[Revision — Agent-Centric Design Principles]

While improving functionality, apply the following design principles to optimize the UI for automated agents. The goal is functionality and testability, not human aesthetics.

### A. Visual Clarity and Simplicity
- Use a simple color scheme (e.g., black text on a white background).
- Avoid decorative elements that do not serve a functional purpose, such as animations, gradients, or shadows.
- Establish a clear visual hierarchy using typography and spacing. Logically group related controls.

### B. Robust Agent Interaction
- All interactive controls must be clearly labeled and sized appropriately to be easily and unambiguously targeted by automation tools.
- Support keyboard-based interaction for all core functionality. Navigable elements should have clear focus indicators.
- Prioritize immediate state updates upon interaction. Avoid complex, multi-step confirmation dialogs for actions where direct manipulation is sufficient.
- All critical functionality should be accessible within a standard 1280x720 viewport without requiring scrolling.

### C. Transparent State Management
- The DOM must serve as a reliable, single source of truth for the application's state.
- Ensure that any significant state change (e.g., a result is generated, a calculation is complete) is clearly and synchronously reflected in the DOM. This can be achieved by updating element attributes, text content, or values.
- Interactive elements and state indicators must have unique and stable IDs to facilitate reliable testing and interaction.

### D. Versatile Input Handling
- For continuous inputs (like sliders), provide alternative discrete control mechanisms (e.g., step buttons, direct text input). No interaction should rely solely on pointer-dragging.
- Input validation should be permissive and should not block inputs that an automated task might

reasonably provide.
- Distinguish between actions that cause immediate, reversible state changes (e.g., selecting an option) and those that trigger irreversible, multi-step processes (e.g., submitting a form).

### E. Behavior Preservation
- Simplifying the visual design must not alter the core interaction logic.
- Any user action that was immediate in initial version must remain immediate in the revised version.

Please generate the complete improved HTML file:

[Revision from Unsupported Tasks]

You are tasked with improving a web application to support additional tasks that are currently unsupported.

## CONTEXT
Application: {app_name}
Model: {model_name}
Total Unsupported Tasks: {len(unsupported_tasks)}
Original HTML Length: {len(initial_html.strip())}

## OUTPUT FORMAT
Generate a single, complete, and self-contained HTML file. The file must be fully functional, including all necessary CSS and JavaScript, from '!DOCTYPE html' to '/html'. Do not use placeholders or truncate the code.

## ORIGINAL INITIAL WEBSITE (FULL)

```
{initial_html}
```
 ## UNSUPPORTED TASKS ANALYSIS
{unsupported_summary}

## CODE PRESERVATION CONTRACT (Non-Regression)
{'' if ablate_no_contract else (non_regression_contract_prompt or '').strip()}

## IMPROVEMENT REQUIREMENTS

### 1. Task Support Issues to Address
Based on the unsupported task analysis, you must ADD missing functionality:
- Add missing DOM elements that tasks expect to exist
- Implement missing JavaScript functionality for user interactions
- Add missing form controls and input handling
- Implement missing display areas and visual feedback
- Add missing navigation and UI components

### 2. Implementation Guidelines
- PRESERVE all existing working functionality from initial version
- ADD new HTML elements with unique IDs for automation
- IMPLEMENT complete JavaScript event handlers for new features
- ENSURE new UI elements are properly styled and visible
- DO NOT introduce new input constraints that would block task inputs implied by tasks; validation must be permissive and must not reduce what the initial version allowed.
- DO NOT auto-trigger flows on load that change initial states (no auto-generation, auto-download, auto-navigation). Start in a neutral, idle state.
- FIT critical controls within a 1280x720 viewport without scrolling. Avoid multi-panel hub layouts and unnecessary panels that push controls below the fold.
- IMPLEMENT missing workflows and user interaction patterns

### 3. DOM Structure Requirements
- Every new interactive element MUST have a unique ID
- New form controls must have proper event listeners
- New content areas must update appropriately on state changes

- New visual feedback must be implemented for added interactions

### 4. JavaScript Functionality Requirements
- All new user interactions mentioned in unsupported tasks must be fully implemented
- New state changes must be reflected in the DOM immediately
- New event handlers must properly update all related UI elements
- Any new game logic, scoring, timing must be complete and functional

## OUTPUT REQUIREMENTS
Generate a complete and fully functional HTML file that:
1. Maintains all existing functionality from initial version.
2. Adds the missing functionality required to support the new tasks.
3. Implements all necessary DOM elements and JavaScript for task support.
4. Ensures all new features are robust and testable.

Commenter JSON (if provided)
- If upstream provides a commenter JSON analysis with 'actionable_changes', follow those changes first, precisely and surgically.

Surgical Revision Policy
- Preserve existing IDs; do not rename or remove working elements from initial version.
- Avoid large rewrites. Patch only the functions, event handlers, and minimal markup necessary to satisfy the failed/unsupported tasks.
- Preserve working logic from initial version; do not regress features that already work.
- Reuse existing elements/IDs for state wherever possible; only add new IDs if strictly necessary to expose the state of new logic.
- Preserve initial version immediacy semantics. Do NOT introduce extra confirmation steps as prerequisites where initial version achieved completion via immediate interactions. Implement functional logic first, then expose proxies from the same code path; never update proxies without the underlying state change.

Please generate the complete improved HTML file:

**CUA Policy Prompt.** Below, we show the prompt used by the CUA Policy.

### CUA Policy Prompt

You are a GUI agent. You are given a task and your action history, with screenshots. You need to perform the next action to complete the task.

## Output Format

```
Thought: ...
Action: ...
```

## Action Space

click(point='x1 y1')
left_double(point='x1 y1')
right_single(point='x1 y1')
drag(start_point='x1 y1', end_point='x2 y2')
hotkey(key='ctrl c') # Split keys with a space and use lowercase. Also, do not use more than 3 keys in one hotkey action.
type(content='xxx') # Use escape characters \', \", and \n in content part to ensure we can parse the content in normal python string format. If you want to submit your input, use \n at the end of content.
scroll(point='x1 y1', direction='down or up or right or left') # Show more information on the 'direction' side.
wait() #Sleep for 5s and take a screenshot to check for any changes.
finished(content='xxx') # Use escape characters \', \", and \n in content part to ensure we can parse the content in normal python string format.

```
## Note
- Use {language} in 'Thought' part.
- Write a small plan and finally summarize your next action (with its target element) in one sentence in
'Thought' part.

## User Instruction
{instruction}
```

**Dashboard Commenter Prompt.** Below, we show the prompt used by the Dashboard Commenter.

---

**Dashboard Commenter Prompt**

You are diagnosing UI design issue that caused a task failure for a Computer-Use Agent (CUA). Your goal is to conduct a root cause analysis based on a core set of design principles and output a structured diagnostic report in JSON format. This report will guide the next iteration of UI code generation.

You will be provided with two images:
1. The current website state (Resolution: {width}x{height})
2. A storyboard summarizing the failed task attempt, arranged as a grid of step screenshots (variable count) fitted into a 1920x1080 canvas

Your analysis must be guided by the following Agent-Centric UI Design Principles:

1. State Visibility: Any significant state change resulting from an agent's action must be clearly and synchronously reflected in the DOM. This can be achieved by updating element attributes, text content, or values. Ambiguous or out-of-band feedback (like temporary toast notifications) is considered a violation.
2. Interaction Robustness: All UI components critical for task completion must be visible and actionable within a standard 1280x720 viewport without requiring scrolling. Elements should have clear, stable identifiers.
3. Input Permissiveness: Input fields and controls should accept the most general data format required for the task, avoiding overly restrictive client-side validation that may block agent inputs.
4. Predictable Behavior: The UI should remain in a stable, neutral state upon loading.

Based on these principles, analyze the provided materials and output a compact JSON object.

Output strictly as JSON with these keys only:
- issues: An array of up to 3 short strings identifying the primary UI problem categories, derived from the violated principles (e.g., "visibility", "interaction", "feedback").
- actionable_changes: An array of 3–6 diagnostic statements. Each statement must identify a specific UI element (referencing selectors/IDs) and explain which design principle it violated, providing a root cause for the failure. Example: "The element '#submit-btn' violates the Interaction Robustness principle, as it is not visible in the default viewport."
- fit_within_screen: A diagnostic boolean flag. Set to 'true' only if the primary reason for failure was a violation of the Interaction Robustness principle concerning viewport visibility.
- avoid_regressions: A confirmation flag, set to 'true', signifying that the diagnosis adheres to a "minimal intervention" philosophy. This confirms the analysis focuses solely on fixing the observed failure without disturbing unrelated, functional parts of the UI.

Respond with JSON only, no extra text.

---

## C  EXPERIMENT DETAILS

For the GPT-5 experiments, we configured GPT-5 with high verbosity and high reasoning effort for coding and low verbosity and minimal reasoning effort for commenting. For the Qwen experiments, we used the Qwen3-Coder-30B-A3B-Instruct model for coding and Qwen2.5-VL-72B-Instruct for commenting. Experiments that involve the CUA policy used UI-TARS-1.5-7B for the UI-TARS CUA and OpenAI Computer-Use-Preview for the Operator CUA. The Task Proposer and the Verifier were both GPT-5 with high verbosity and high reasoning effort. In the CUA policy test we set the

maximum steps to 20 to prevent infinite loops and we ran the environment with Playwright. The CUA performed coordinate-based Computer Use actions only and did not interact with elements directly. This design makes the evaluation more challenging and more informative for UI design, since element layout and visibility become critical.

**Effects by Refinement Round.** As shown in the figure 8, iteratively applying revision can consistently bring improvements on the function completeness for all coders. Interestingly, it can be observed that the CUA success rate of GPT-5 coder may drop after repeated revision, while Qwen3-Coder-30B and GPT-4o can consistently gain from repeated revision. This indicates that the revision improvement may saturate for strong coders, but weak coders can be improved with iterative feedback and revisions.

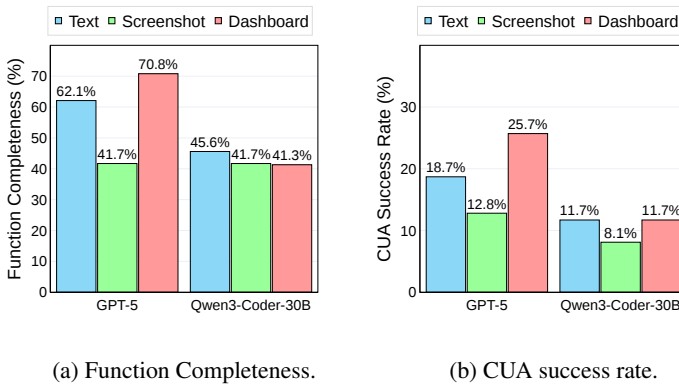

(a) Function Completeness.   (b) CUA success rate.

Figure 6: Ablation comparison of different commenters across two evaluation dimensions.

Figure 6 demonstrates the results when using different types of commenters for revision based on CUA navigation feedback. Besides dashboard commenter, there are two additional commenters which either use text only or screenshot only for CUA policy trajectory information. Text-only commenter will only percept the same text information used in the dashboard, *i.e.,* task description, task expected outcome, CUA thought and action, etc. Screenshot-only commenter will only percept the same screenshots used in the dashboard. Besides the CUA trajectory information, all the three commenters will see the full UI screenshot for UI analysis. It can be observed that dashboard can bring significant improvements for GPT-5 commenter and coder, but its performance is slightly lower than text-only commenter for Qwen commenter and coder. This indicates that dashboard offers comprehensive visual and textual cues for revision, which in turn requires strong visual perception to fully capture the information; meanwhile, a text-only commenter remains a reliable proxy for commenters with weaker perception.

## D    FULL STATISTICS AND EXAMPLES

Below, we display the full statistics and corresponding examples.

## E    QUALITATIVE ANALYSIS

In this section, we present four representative revision cases—*artisan-csa*, *color-match-challenge*, *csv-to-charts*, and *festival-lights-show*. Each row displays the initial UI alongside its revised versions, evaluated under two criteria: Function Test and CUA Test. Across the four cases, the revisions demonstrate distinct patterns of improvement. Revisions based on the Function Test, which addresses unsupported tasks, tend to focus on adding underlying functionality, sometimes with subtle visual changes. For example, the *festival-lights-show* revision added a crucial "Running" state indicator, and the *csv-to-charts* revision added a button to select a delimiter. In contrast, revisions based on the CUA Test consistently yield more significant visual modifications geared towards agent accessibility. For most websites, this meant adding buttons with clear boundaries and visual hints.

Table 7: **Distribution and examples of six domains in AUI-Gym.** For each domain, we show a website created by GPT-5, paired with 30 tasks (blue) simulating real-world usage. Each task is further linked to a rule-based verifier (green).

| Domain | #Apps | Percen-tage | Example Instruction | GUI created by GPT-5 |
|---|---|---|---|---|
| App | 11 | 21% | Create a single-page app in a single HTML file with the following requirements: 
 - Name: Healthy Meal Tracker 
 - Goal: Log meals and nutrition info. 
 - Features: Ingredient list, calories per meal, daily summary. 
 - The UI should be clean with food icons. 
 **Task: Add five meals for today's date (any names/ingredients) so today's meal count reaches at least 5.** 
 **Rule: #dailyMealCount >= 5** | |
| Landing | 10 | 19% | Create a single-page app in a single HTML file with the following requirements: 
 - Name: Nonprofit Impact Report 
 - Goal: Show measurable results of programs. 
 - Features: Infographics, success stories, donation link. 
 - The UI should be inspiring and visually engaging. 
 **Task: Navigate to Success Stories and expand the first story card to reveal the full narrative.** 
 **Rule: #slides .slide:first-child button[aria-expanded] == 'true' OR #slides .slide:first-child.expanded exists** | |
| Game | 9 | 17% | Create a single-page app in a single HTML file with the following requirements: 
 - Name: Typing Rain 
 - Goal: Type falling words before they reach the bottom. 
 - Features: Increasing difficulty, accuracy tracker, score. 
 - The UI should be the city background with animated raindrop words. 
 **Task: In a single run, achieve a score of at least 500 points.** 
 **Rule: #scoreValue >= 500** | |
| Interactive | 9 | 17% | Create a single-page app in a single HTML file with the following requirements: 
 - Name: Festival Lights Show 
 - Goal: Control a virtual light show. 
 - Features: Color changes, patterns, music sync. 
 - The UI should be vibrant and dynamic. 
 **Task: Enable Music Sync, start playback, then pause the built-in track; confirm audio status is Paused while Music Sync remains enabled.** 
 **Rule: #audioStatus == 'Paused' AND #syncBadge != 'Sync: Off'** | |
| Tool | 7 | 13% | Create a single-page app in a single HTML file with the following requirements: 
 - Name: Customer Journey Flow 
 - Goal: Sketch customer journey stages and connections. 
 - Features: Add/edit stages, connect nodes, view JSON of the flow. 
 - The UI should be simple and full-screen. 
 **Task: Create 'Social Ad' and 'Search Ad' leading to 'Landing Page', then to 'Consideration' and 'Purchase' (two branches merging into one path).** 
 **Rule: #io-json contains 'Social Ad' AND #io-json contains 'Search Ad' AND #io-json contains 'Landing Page' AND #io-json contains 'Consideration' AND #io-json contains 'Purchase'** | |
| Utility | 6 | 12% | Create a single-page app in a single HTML file with the following requirements: 
 - Name: Pomodoro 
 - Goal: Time focus and break sessions. 
 - Features: Focus/break modes, timers, basic controls. 
 - The UI should be minimal and distraction-free. 
 **Task: Start a short break and verify the mode label and starting time show a 5-minute break.** 
 **Rule: #lblSession == 'Short Break' AND #lblTime contains '05:00'** | |

In both *color-match-challenge* and *csv-to-charts*, both revision types improved accessibility by presenting more information and controls upfront, reducing the need for scrolling. A key CUA-friendly

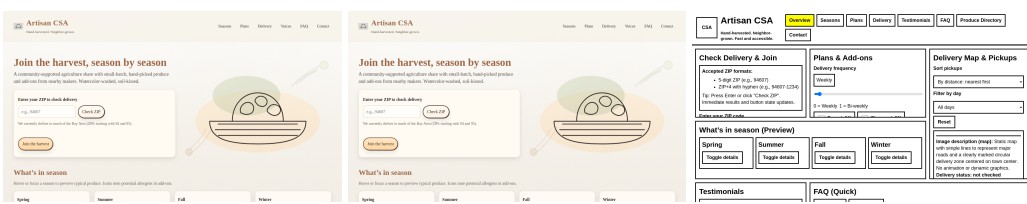

|  | | |
| :---: | :---: | :---: |
| Initial UI | w. Task Solvability Feedback | w. CUA Navigation Feedback |

(a) `artisan-csa`: Create a single-page app, in a single HTML file, for a community-supported agriculture program with a hand-drawn, watercolor aesthetic.

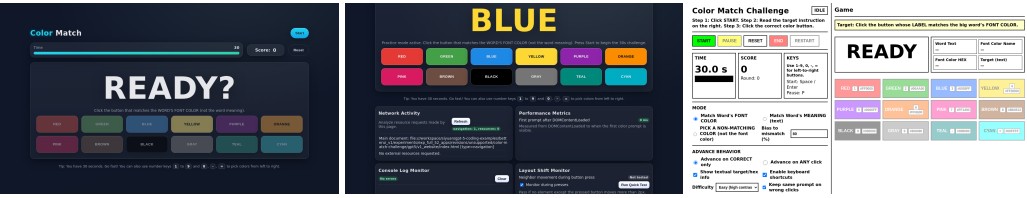

|  | | |
| :---: | :---: | :---: |
| Initial UI | w. Task Solvability Feedback | w. CUA Navigation Feedback |

(b) `color-match-challenge`: Create a single-page app in a single HTML file for a fast-paced "color match" game. - Show a word (e.g., "RED") in a random font color — player must click the correct color button (not the word meaning). - Keep score based on correct answers within 30 seconds. - Use large typography, color-coded buttons, and smooth button press animations.

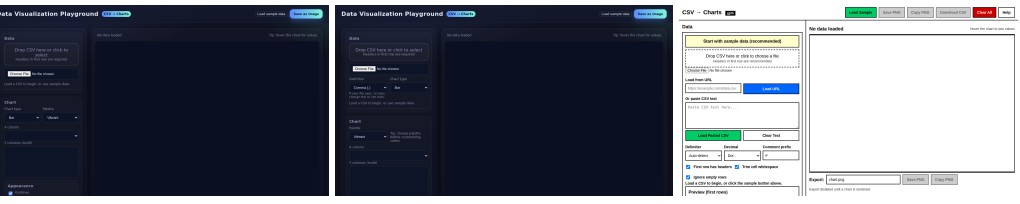

|  | | |
| :---: | :---: | :---: |
| Initial UI | w. Task Solvability Feedback | w. CUA Navigation Feedback |

(c) `csv-to-charts`: Create a single-page app in a single HTML file with the following requirements: - Name: Data Visualization Playground - Goal: Upload CSV and generate charts. - Features: Chart type selector, color customization, save as image. - The UI should be modern with a focus on charts.

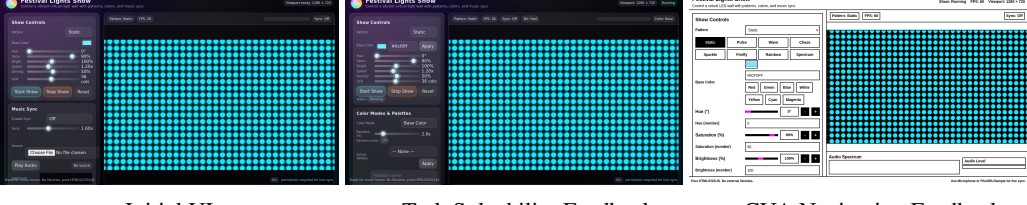

|  | | |
| :---: | :---: | :---: |
| Initial UI | w. Task Solvability Feedback | w. CUA Navigation Feedback |

(d) `festival-lights-show`: Create a single-page app in a single HTML file with the following requirements: - Name: Festival Lights Show - Goal: Control a virtual light show. - Features: Color changes, patterns, music sync. - The UI should be vibrant and dynamic.

Figure 7: Qualitative comparison of UI *v.s.* AUI. Each row shows an initial UI (left), its revision based on function (middle), and its revision based on CUA-failed tasks (right).

adaptation is seen in *festival-lights-show*, where "increase" and "reduce" buttons were added as a complement to sliders, providing a more direct and reliable interaction method for automated agents.

## F    THE USE OF LARGE LANGUAGE MODELS

We used large language models (LLMs) in two limited ways. First, during manuscript preparation, LLMs were employed solely for surface-level editing (*e.g.,* grammar correction and minor rephrasing) to improve readability; they were not used to generate research ideas, methods, experiments, or conclusions. Second, in our benchmark experiments, LLMs were included as baseline models for

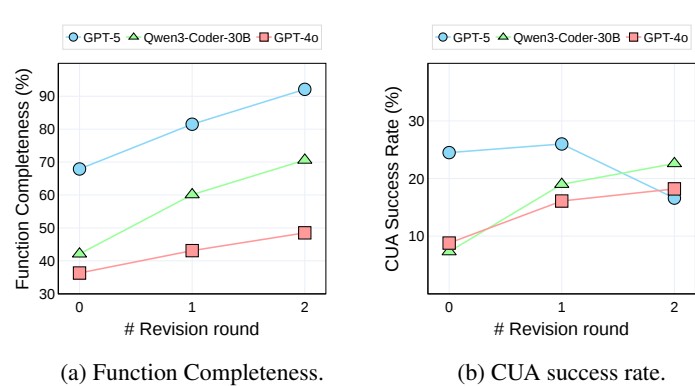

(a) Function Completeness.

(b) CUA success rate.

Figure 8: Effects of Revision round.

comparison, with results reported transparently in the main paper. All core research contributions, dataset design, and analyses are the sole work of the authors.

