# OpenReview forum: "Computer-Use Agents as Judges for Automatic GUI Design"
_ICLR.cc/2026/Conference — ICLR 2026 Conference Withdrawn Submission_

### Official Review · Reviewer_AV2r · 2025-10-27

**Soundness:** 2
**Presentation:** 2
**Contribution:** 4
**Rating:** 4
**Confidence:** 4

**Summary:**

This paper proposes a new collaborative design framework with the Coder acting as the website designer and the CUA serving as the judge, aiming to improve CUA performance by designing environments tailored to it. The designer creates a website defined by a task, the CUA model navigates the site, and the environment collects feedback to help the designer refine the website. The authors introduce a testbed called AUI-Gym, which includes diverse applications and domains to comprehensively cover website-design tasks; using this testbed, they evaluate several Coder and CUA models with functional completeness and success rate as metrics. They also employ a verifier to extract rule-based metrics for assessing the feasibility of generated websites and compress navigation trajectories into a single screenshot to preserve detailed feedback while substantially reducing token costs. Their evaluation suggests that task solvability is foundational and can be improved via failure-driven functional summarization, whereas CUA navigation remains the primary bottleneck.

**Strengths:**

1. The proposed framework is highly innovative and promising: the authors are guided by the principle that GUIs designed for humans may not suit autonomous agents, and they propose a collaborative framework that enables the Coder and the CUA to co-evolve. This is one of the earliest efforts to conceptualize the idea.
2. They provide clear instructions on how to build a testbed for the framework and how to evaluate its performance, offering valuable insights for future work.

**Weaknesses:**

While the idea and concept are brilliant and exciting, the benchmark design and the paper’s presentation are weak. The authors do not clearly explain the feasibility or justify many of their design choices. For example, they claim that AUI‑Gym is human‑free, yet Figure 3 shows humans providing input queries and performing filtering and verification. They use GPT‑5 as a functional checker for generated websites but do not address possible defects in rules produced by LLMs. The paper also contains unclear expressions—for instance, it does not state which CUA model is used in the main table.

Suggested corrections for the typos you noted: Line 73, duplicate text that may be due to a rush; Line 144: Using "more recently" is inappropriate because these papers were published earlier than the previously cited works.

**Questions:**

1. What proportion of human effort went into creating AUI-Gym?

2. Regarding Section 3.3, how can we trust GPT‑5’s performance? How well do the verification rules produced by GPT‑5 agree with those created by human designers? Are all of the rules correct and complete?

3. How would the authors respond to the concern that, although automatically generated websites are intended to improve agent performance, they may oversimplify real websites and thus differ substantially from real‑world sites?

---

### Official Review · Reviewer_a1h8 · 2025-10-29

**Soundness:** 2
**Presentation:** 2
**Contribution:** 2
**Rating:** 4
**Confidence:** 4

**Summary:**

The paper introduces a benchmark and framework for automatic graphical user interface (GUI) design in which a language model generates interfaces and an autonomous agent tests them to provide feedback. The system integrates two feedback types—functional solvability and navigation performance—to iteratively refine the interface through an automated loop. A dataset of web applications and rule-based task verifiers enables evaluation without human supervision. Experiments show that iterative feedback improves interface functionality and agent task success, demonstrating the feasibility of agent-driven interface design.

**Strengths:**

1. Presents a clear conceptual framework that redefines GUI design as an agent–environment co-adaptation process rather than a static, human-centered task.

2. Proposes a systematic feedback mechanism combining functional and navigation signals, offering a structured way to refine UIs iteratively.

3. Demonstrates that automated feedback loops can improve interface functionality, suggesting potential for scaling agent-centric design pipelines.

**Weaknesses:**

1. The framework’s improvements rely on feedback from specific computer-use agents, effectively tuning the coder to each agent’s perceptual biases rather than achieving generalizable interface quality.

2. The evaluation uses synthetic, single-page applications and templated tasks that lack the complexity and variability of real-world GUIs.

3. The paper assumes reliable grounding and interaction accuracy from the agents but provides no validation or error analysis of mis-clicks or perception failures.

4. The iterative design loop has no convergence analysis or discussion of failure cases, leaving unclear whether revisions can oscillate or stall.

**Questions:**

1. Failure and convergence behavior – How stable is the iterative Coder–CUA loop in practice? Are there cases where the process stalls, oscillates, or repeatedly fails to resolve the same issue? Quantitative or qualitative analysis of non-convergent cases would clarify the robustness of the framework.

2. Grounding accuracy and attribution of failure – How do the authors distinguish genuine interface design flaws from CUA perception or grounding errors (e.g., mis-clicks or missed buttons)? Have they quantified how often failures are due to agent limitations rather than UI design?

3. Design bias induced by CUA feedback – Do the authors agree that the feedback process effectively guides the coder to generate layouts that align with the CUA’s perceptual biases (e.g., favoring large, high-contrast, centered elements)? If so, how might this affect generalization to other CUAs with different visual grounding behaviors?

4. Trade-off between human and agent usability – When simplifying or de-stylizing UIs to improve agent navigation, did the authors evaluate whether such changes reduce human readability or aesthetic quality? Discussing this trade-off could clarify the broader design implications of agent-centric optimization.

---

### Official Review · Reviewer_JjAX · 2025-10-31

**Soundness:** 1
**Presentation:** 2
**Contribution:** 1
**Rating:** 2
**Confidence:** 4

**Summary:**

The paper introduces AUI-Gym, a new benchmark for Automatic GUI development using Computer-Use Agents (CUA), and a Coder-CUA in Collaboration framework with an integrated Dashboard that produces concise visual summaries. It is found that SOTA Coders can generate GUIs that look good on the surface but that still have limitations, such as low usability and broken navigation capabilities. Ultimately, the paper argues to "design for agents" rather than "merely adapting human-centric interfaces".

**Strengths:**

The paper represents solid engineering work, with interesting practical results.

The paper promises to release everything needed to ensure full replication, which is always a plus.

**Weaknesses:**

There are quite a few issues with the paper: framing, contribution, and scholarship. I also provide a few minor comments for improvement below.

FRAMING

The paper makes assumptions that are not supported by evidence. For example, while it is true that "GUIs are designed primarily for humans", the fundamental question here is: Why should agents operate in the same environment? The paper says that "stylistic details crucial for humans are redundant for agents" so Why can't agents work e.g. with the Document Object Model only? Actually, agents could leverage the same visual and textual cues that help users scan webpages (cf. saliency effects) or read documents (cf. kerning, line-height, etc.). Unfortunately the paper does not discuss these points, which results in a weak framing of the work.

CONTRIBUTION

It looks like the paper tries to claims too many things, which eventually makes it a bit difficult to follow. AUI-Gym, Coder-CUA collaboration framework, CUA Dashboard "that produces concise visual summaries"... Any of these items could even be a paper by itself. I do not think that "evaluation insights" should be considered a contribution, by the way, given that evaluations are a means to an end (e.g. "prove that task XYZ can be solved faster/better/etc. with system A vs B vs C").

On the other hand, it is unclear how this paper contributes to ICLR, since it is mostly about engineering (an otherwise compeling system) instead of learning representations.

SCHOLARSHIP

The paper misses many relevant papers, such as:
* Agent-as-a-Judge: Evaluate Agents with Agents https://arxiv.org/abs/2410.10934
* Large Language Model-Brained GUI Agents: A Survey https://arxiv.org/abs/2411.18279
* GUI Agents: A Survey https://arxiv.org/abs/2412.13501
* Judge: Effective State Abstraction for Guiding Automated Web GUI Testing https://dl.acm.org/doi/abs/10.1145/3736162

Without a proper positioning in the research literature, it is a bit difficult to understand the novelty of this paper.

MINOR COMMENTS

The AUI-Gym pipeline followed a Human-in-the-Loop approach but no much details are given about the input sources. For example, it is unclear if the gpt-5-coding-examples repository has more app examples or task prompts that could have been considered.

The citation style should be improved: Check the use of `\citep` instead of `\cite`.

The Verifier "algorithm" in Section 3.3 is not informative, so it can be removed.

The title in the references should have proper casing; e.g. "GPT-4" instead of "Gpt-4", "GUI agents" instead of "gui agents", "UI layout" instead of "Ui layout", "UI grammar" instead of "ui grammar", etc.

**Questions:**

Please see my review above.

---

### Note · Authors · 2025-11-24

I have read and agree with the venue's withdrawal policy on behalf of myself and my co-authors.